# Screening and Identification of a *Streptomyces* Strain with Quorum-Sensing Inhibitory Activity and Effect of the Crude Extracts on Virulence Factors of *Pseudomonas aeruginosa*

**DOI:** 10.3390/microorganisms11082079

**Published:** 2023-08-13

**Authors:** Zhidong Zhang, Yang Sun, Yuanyang Yi, Xiaoyu Bai, Liying Zhu, Jing Zhu, Meiying Gu, Yanlei Zhu, Ling Jiang

**Affiliations:** 1Xinjiang Key Laboratory of Special Environmental Microbiology, Institute of Microbiology, Xinjiang Academy of Agricultural Sciences, Urumqi 830091, China; zhangzheedong@sohu.com (Z.Z.); 15022970169@163.com (Y.Y.); zhujing2020@hotmail.com (J.Z.);; 2College of Life Sciences, Xinjiang Normal University, Urumqi 830054, China; zhuyanlei1226@163.com; 3College of Food Science and Light Industry, Nanjing Tech University, Nanjing 211816, China; sunyang@njtech.edu.cn; 4College of Chemical and Molecular Engineering, Nanjing Tech University, Nanjing 211816, China

**Keywords:** quorum-sensing inhibitors, Actinomycete, *Pseudomonas aeruginosa*, virulence factors

## Abstract

Quorum-sensing (QS) is involved in numerous physiological processes in bacteria, such as biofilm formation, sporulation, and virulence formation. Therefore, the search for new quorum-sensing inhibitors (QSI) is a promising strategy that opens up a new perspective for controlling QS-mediated bacterial pathogens. To explore new QSIs, a strain named *Streptomyces* sp. D67 with QS inhibitory activity was isolated from the soil of the arid zone around the Kumutag Desert in Xinjiang. Phylogenetic analyses demonstrated that strain D67 shared the highest similarity with *Streptomyces ardesiacus* NBRC 15402T (98.39%), which indicated it represented a potential novel species in the *Streptomyces* genus. The fermentation crude extracts of strain D67 can effectively reduce the violacein production produced by *Chromobacterium violaceum* CV026 and the swarming and swimming abilities of *Pseudomonas aeruginosa*. It also has significant inhibitory activity on the production of virulence factors such as biofilm, pyocyanin, and rhamnolipids of *P. aeruginosa* in a significant concentration-dependent manner, but not on protease activity. A total of 618 compounds were identified from the fermentation crude extracts of strain D67 by LC-MS, and 19 compounds with significant QS inhibitory activity were observed. Overall, the strain with QS inhibitory activity was screened from Kumutag Desert in Xinjiang for the first time, which provided a basis for further research and development of new QSI.

## 1. Introduction

Quorum sensing inhibitor (QSI) is a chemical that can block the quorum-sensing pathway [1]. Previous studies have shown that quorum-sensing (QS) is involved in many important biological processes, such as spore formation, virulence, and biofilm formation [2], and can cause various diseases in plants and animals and food spoilage [3,4]. QSI can block the QS pathway and reduce pathogen resistance, virulence factor metabolism, etc., by inhibiting the synthesis enzymes of QS signal molecule synthase or producing signal molecular analogs to competitively bind to transcription protein receptors without causing significant survival pressure on pathogens, providing new ideas for pathogen control and the development of related new drugs [5]. Therefore, the screening of safe and highly efficient QSI has become one of the research hotspots in the fields of human and animal health, agricultural plant protection, and food preservation. *Pseudomonas aeruginosa* (*P. aeruginosa*) is a typical gram-negative pathogenic bacterium that is widely distributed in various ecological environments and is capable of causing a variety of infectious diseases. It is also a member of multiple and pan-antibiotic-resistant bacteria [6]. Available studies have shown that the metabolic production of virulence factors of *P. aeruginosa* such as biofilm, pyocyanin, rhamnolipids, proteases, and their motility are associated with QS systems, such as the LasI/LasR signaling system that mainly regulates the expression of some proteases, the RhlI/RhlR system that regulates the production of rhamnolipid, pyocyanin, cyanide, lipase, alkaline protease, and elastase, and the pseudomonas quinolone signal system can regulate the production of rhamnolipids [7], and when its QS pathway is inhibited, the corresponding metabolic capacity is reduced. Therefore, the use of QSI provides a new idea for the control of *P. aeruginosa* [8].

Currently, the commonly used QSI screening model, *Chromobacterium violaceum* CV026 (*C. violaceum*), is to mutate the mini-Tn5 gene of *C. violaceum* ATCC 31532 by molecular means, resulting in its own non-production of N-acyl-homoserine lactones (AHLs) signaling molecules, and when exogenous signaling molecules, AHLs, are artificially added, the strain produces purple pigment. Whereas, when a QSI is added, it blocks or disrupts the signaling molecule transmission pathway but does not inhibit the growth of *C. violaceum* CV026, resulting in a fuzzy, transparent circle that does not contain purple pigment [9]. *P. aeruginosa* has a similar QS signaling molecule to that of *C. violaceum* CV026, and the screening of this model can be effective for the screening of *P. aeruginosa* QSI. At present, there are more reports on *P. aeruginosa* QSI, mostly from plants and microorganisms, such as extracts from plants such as *Andrographis paniculata*, *Callistemon viminalis*, and *Hypericum monogynum* [10,11,12]; QSI from microbial sources such as *Streptomyces*, *Bacillus*, and *Arthrobacter* have also been reported, especially the special environment of Actinomyces, which has become one of the focuses of QSI mining [13,14,15].

In this study, a group of actinomycetes was isolated from the soil of the arid zone around Kumutag Desert, and a *Streptomyces* sp. named D67 with strong group induction activity was identified through the “*C. violaceum* CV026 screening model”, and its fermentation products were extracted to examine the effects of the crude extract on the swarming and swimming of *P. aeruginosa* and the inhibition of biofilm, pyocyanin, rhamnolipid, and protease production, which laid a solid foundation for the molecular analysis and application of the group inhibition activity of strain D67.

## 2. Materials and Methods

### 2.1. Bacterial Strains and Growth Conditions

*Streptomyces* sp. D67 was isolated from the Kumutag Desert. Soil samples were collected in July 2021 at the eastern edge of the Kumutag Desert, Xinjiang. Five sampling sites, about 5 km away from each other, were selected within a diameter of 20 km. Three samples (1 kg soil for each sample) were collected randomly within a 100 × 100 m area at each site, and each sample was collected at a depth of 10–15 cm from the surface using the five-point serpentine sampling method. The soil sample (2 kg) was mixed well at each site after removing impurities obtained by the quartering method [16] and stored at 4 °C. Subsequently, actinomycetes were isolated and purified from the soil samples. *C. violaceum* CV026 was kindly provided by Prof. Wang Yan (School of Marine Life, Ocean University of China, Qingdao, China), and *P. aeruginosa* CMCC(B)10104 was provided by the Xinjiang Microbial Resources Conservation and Management Center, Urumqi, China.

### 2.2. Screening of QS-Inhibiting Active Actinomycetes

The fresh actinomycete strain to be tested was inoculated into a 250 mL conical flask containing 50 mL of GAUSE No.1 medium and incubated for 7 d at 30 °C with constant shaking at 150 r/min. The fermentation broth was filtered through a 0.22 μm membrane, and the filtrate was collected and stored at −20 °C.

The *C. violaceum* CV026 was grown in Luria-Bertani (LB) medium at 30 °C and shaken at 150 r/min for 16 h. 1 mL of CV026 and 20 μL C6-HSL (100 μmol/mL) were added into a petri dish, then 25 mL of LB solid medium (50 °C) was added, mixed quickly, and put into an Oxford cup (8 mm outer diameter), and the medium was allowed to solidify and set aside.

Take 100 μL of fermentation filtrate into the wells and use sterile water as a control; incubate at a positive constant temperature of 30 °C for 24 h. Observe the discoloration of the plates and record the strains with fuzzy, transparent circles.

### 2.3. Morphological Observation and Molecular Identification of Strains

The strains were selected on solid medium plates with sterilized coverslips and incubated at 30 °C for 4 d. The colony size, color, and spore morphology were observed. The genomic DNA of the strain was extracted using the TIANamp Bacteria DNA Kit (Tiangen Biochemical, Beijing, China), and PCR was performed using the bacterial 16S rRNA gene sequence universal primers 27F (5′-AGAGTTTGATCCTGGCTCAG-3′) and 1492R (5′-GGTTACCTTGTTACGACTT-3′). The PCR amplification products were detected by agarose gel electrophoresis, purified, and sequenced by Sangon Biotech (Sangon Biotech, Shanghai, China). The 16S rRNA gene sequences of the strains were uploaded to the NCBI database for comparison, the sequences of related similarity pattern strains were retrieved, and the Neighbor-Joining phylogenetic tree was constructed using MEGA version 7 software to determine their taxonomic status.

### 2.4. Preparation and Functional Validation of Crude Extract

The fermentation broth was centrifuged, and the supernatant was extracted with an equal volume of ethyl acetate for 3 h. The organic phase was evaporated at 40 °C in a vacuum rotary evaporator and weighed to obtain the crude extract. The crude extract (150 mg) was weighed and dissolved in dimethyl sulfoxide, filtered through an organic solvent membrane, and the filtrate was stored at −20 °C and set aside.

The QS activity of the crude extract was verified by thin-layer chromatography (TLC)-bioautography [17]. The crude extract (5 μL) was spotted on an aluminum thin layer plate, and the methanol-water system was chosen to unfold the agent system (60:40, *v*/*v*), and the chromatography was performed under airtight conditions, placed on a UV lamp at 254 nm to observe the fluorescent spots. Furthermore, LB agar mixed with *C. violaceum* CV026 and C6-HSL was spread on the TLC plate, solidified, and incubated at 30 °C for 36 h. The discoloration of the plate was observed.

### 2.5. Effect of Crude Extract on Bacterial Inhibition

The 96-well plate was inoculated with *C. violaceum* CV026 and *P. aeruginosa* (middle and late logarithmic phases) at 2% inoculum in LB liquid medium containing different concentrations (0.005, 0.05, 0.5, 1, 2, 5, 10, and 20 mg/mL) of crude extracts; the same volume of dimethyl sulfoxide was used as the negative control and the same volume of sterile water as the positive control. The mixtures were incubated at 30 °C for 24 h with shaking, and the OD590 values were measured every 2 h to determine the bacterial inhibition of the crude extracts.

### 2.6. Effect of Crude Extract on the Production of Purple Pigment by C. violaceum

The 96-well plate was used to measure the production of purple pigment. Briefly, 120 μL of LB liquid medium (containing 0.1 μmol C6-HSL) was added to the wells, and the *C. violaceum* CV026 (middle and late logarithmic phase) were inoculated with 2% inoculum, and appropriate amounts of crude extracts were added, and the final concentrations were 0.005, 0.05, 0.5, 1, and 2 mg/mL, respectively. The mixtures were incubated for 24 h at 30 °C with shaking in a microplate reader, and the same volume of dimethyl sulfoxide was used as the control. The method with reference to Reimann et al. was slightly modified [18]. Briefly, 100 μL of the above fermentation broth was centrifuged at 10,000 r/min for 5 min at room temperature, the supernatant was discarded, 100 μL of DMSO was added and mixed to make the purple pigment of *C. violaceum* CV026 fully dissolved, the supernatant was centrifuged, the OD600 value was determined by the microplate reader, and the effect of the crude extract on the production of purple pigment by *C. violaceum* CV026 was calculated.

### 2.7. Effect of Crude Extracts on the Swarming and Swimming Properties of P. aeruginosa

The method concerning Kapadia et al. was slightly modified [19]. Briefly, swimming agar plates (peptone 10.0 g/L, glucose 5.0 g/L, NaCl 5.0 g/L, agar 5.0 g/L, pH natural) and swarming agar plates (tryptone 10.0 g/L, NaCl 3.0 g/L, agar 5.0 g/L, pH natural) containing 5 mg/mL crude extract were prepared, respectively, and equal volume dimethyl sulfoxide was used as a control. *P. aeruginosa* was spotted in the middle of the plates and incubated at 30 °C for 24 h at a constant orthostatic temperature to observe the swarming and swimming of the strains.

### 2.8. Effect of Crude Extract on the Biofilm of P. aeruginosa

A 96-well plate method was used to take 120 μL LB liquid culture based on wells, respectively, to access *P. aeruginosa* solution at 2% inoculum, add a certain amount of crude extract to the final concentration of 0.5, 1, 2, 5 mg/mL, and add an equal amount of dimethyl sulfoxide as a control, and incubate at 30 °C for 24 h in an enzyme marker. The biofilm was obtained by rinsing three times, being naturally air-dried and stained with 1% crystalline violet for 15 min, rinsed and dried, and eluted with 95% ethanol, and the OD590 values were determined and the results were counted [20].

### 2.9. Effect of Crude Extracts on P. aeruginosa Pyocyanin

The method with reference to Miao et al. was slightly modified [21]. Briefly, 100 μL of the bacterial solution (the same as item 2.8) was centrifuged at 8000 r/min for 5 min, the supernatant was extracted, 100 μL of chloroform was added to extract pyocyanin by concussion, and 100 μL of HCl (0.2 mol/L) was used to reverse extract the pyocyanin in chloroform, and when the HCl solution layer changed from colorless to red, the OD520 value was measured, and the results were counted.

### 2.10. Determination of Crude Extracts on P. aeruginosa rhamnosus Lipids

The method with reference to Lou et al. was slightly modified [22]. Briefly, 100 μL of supernatant was obtained by centrifuging the bacterial solution (the same as item 2.8), extracted twice by adding ethyl acetate, evaporated at 40 °C, dissolved by adding 100 μL of ultrapure water, reacted by adding 300 μL of a 2% anthrone-sulfuric acid solution in boiling water for 15 min, and cooled. The blank control group was established and repeated three times, and the OD625 values of the samples and the blank control group were measured and the results were counted.

### 2.11. Determination of Total Protease Production

50 μL of supernatant (the same as item 2.8) was added after punching a 6 mm diameter hole in the center of the protease agar plates (skim milk powder 20.0 g/L, agar 15.0 g/L, pH natural), and incubated at 30 °C for 8 h. The size of the transparent circle was observed and measured.

### 2.12. LC-MS Assay and Validation of Quorum Sensing Inhibitory Activity of Compounds

The fermentation crude extract of *Streptomyces* sp. D67 was sent to BioNovoGene Co., Ltd. for LC-MS detection. The chromatographic conditions: positive ion mobile phases were 0.1% formic acid acetonitrile (B2) and 0.1% formic acid water (A2), and the gradient elution was 2% B2, 0–1 min; 2–50% B2, 1–9 min; 50–98% B2, 9–12 min; 98% B2, 12–13.5 min; 98–2% B2, 13.5–14 min; 2% B2, 14–20 min. 98% B2, 12–13.5 min; 98–2% B2, 13.5–14 min; 2% B2, 14–20 min. The negative ion mobile phase was acetonitrile (B3) and 5 mM ammonium formate water (A3), and the gradient elution was 2% B3, 0–1 min; 2–50% B3, 1–9 min; 50–98% B3, 9–12 min. 98% B3, 12–13.5 min; 98–2% B3, 13.5–14 min; 2% B3, 14–17 min. flow rate: 0.25 mL/min, column temperature: 40 °C, injection volume: 2 μL [23]. 

Main conditions of the mass spectrometry: an electrospray ionization (ESI) source was used with a positive ionization spray voltage of 3.50 kV and a negative ionization spray voltage of −2.50 kV and a capillary temperature of 325 °C A primary scan was performed at a resolution of 70,000, with a primary ion scan range of 100~1000 m/z, and secondary cleavage was performed using a high energy collision dissociation (HCD) operating mode with a collision energy of 30 eV, and a secondary resolution of 17,500 [24].

Some potentially functional compounds were selected through preliminary screening, and the compound standards were prepared at a concentration of 10 mg/ml for functional verification with item 2.2. The standard solution (100 μL) was added to the perforation of the *C. violaceum* CV026 model screening medium, sterile water was used as the control, and the plates were incubated at a positive constant temperature at 30 °C for 24 h. The discoloration of the plates was observed, and the diameter of the transparent circle was recorded.

## 3. Results

### 3.1. Screening and Identification of Quorum-Sensing Inhibitory Active Strains

We screened all kinds of microorganisms in the sample through different culture media, and then, through preliminary fermentation, we used the *Chromobacterium violaceum* CV026(*C. violaceum* CV026) screening model for primary screening and re-screening and finally determined the experimental strain. The Actinomycetes from the Kumutag Desert were screened using the QSI screening model (*C. violaceum* CV026) and a *Streptomyces* sp. named D67 with significant colony inhibition activity was obtained. As shown in Figure 1A, the fermentation broth of this strain could produce a larger transparent circle when compared with the control. Further analysis showed that the fermentation broth did not significantly inhibit the growth of *C. violaceum* CV026, but could significantly inhibit the production of purple pigment by *C. violaceum* CV026. It still had an obvious transparent circle after dilution four times, which proved that this strain has an obvious QS inhibition effect.

Strain D67 can grow well on GAUSE No.1 medium, Bennett medium, ISP2 medium, etc. The colony size was 0.2 cm–0.6 cm × 0.2 cm–0.6 cm when cultured on the GAUSE No.1 agar plates at 30 °C, and it was tightly bound to the medium with a dense texture. Strain D67 formed a white, fluffy aerial mycelium, white spores, and yellowish basal mycelium (Figure 1B,C). The 16S rRNA gene sequence of strain D67 was uploaded to the NCBI for sequence alignment, and a phylogenetic tree was constructed (Figure 2). The sequence of the 16S rRNA gene of strain D67 belonged to the *Streptomyces* genus, and its 16S rRNA gene sequence had the highest similarity (98.39%) with *Streptomyces ardesiacus* NBRC 15402T, followed by *Streptomyces heliomycini* NBRC 15899T (98.11%), which indicated it represented a potential novel species in *Streptomyces*, tentatively named *Streptomyces* sp. D67, and its 16S rRNA gene sequence registration number is ON815269.

### 3.2. Validation of the Effect of Crude Extract on Quorum Sensing

In order to verify the effect of the crude extract on QS, a TLC assay, a bacteriostatic experiment, and a *C. violaceum* CV026 purple pigment production assay were performed, respectively. A crude extract was prepared at a concentration of 150 mg/mL and analyzed by TLC. The results are shown in Figure 3; a clear dark spot under 254 nm UV irradiation with an Rf value of 0.77 was observed. The result of TLC-bioautography showed that there was an obvious strain growing above the thin plate, but the strain did not accumulate the transparent circle of purple pigment, which indicated that the crude extract contained quorum-sensing inhibition activity. In addition, a distinct coloring band on the TLC plate was visualized under short-wavelength UV (254 nm), indicating that the main active group has a 254 nm UV absorption ability (Figure 3B).

Furthermore, the inhibitory effect of crude extract in different concentrations on *C. violaceum* CV026 and *P. aeruginosa* was performed. The results are shown in Table 1. The crude extract at high concentrations significantly inhibited the growth of both *C. violaceum* CV026 and *P. aeruginosa*, but the inhibition effect of the crude extract at low concentrations was different. The growth of *C. violaceum* CV026 was significantly affected at a concentration of 5 mg/mL in the crude extract, while *P. aeruginosa* still grew weakly at 10 mg/mL.

The crude extract (0.005 mg/mL, 0.05mg/mL, 0.5 mg/mL, 1 mg/mL, and 2 mg/mL), which had no obvious inhibition on the growth of *C. violaceum* CV026, was co-cultured with *C. violaceum* CV026, and the pigment production of *C. violaceum* CV026 was detected with the addition of C6-HSL. The results showed that the crude extract had a significant effect on the purple pigment production of *C. violaceum* CV026 in all experimental groups, and the purple pigment production decreased significantly with the increase in the crude extract concentration (Figure 4), indicating that the QSI in the crude extract could significantly inhibit the purple pigment production of the strain, even if the concentration of 0.005 mg/mL of the crude extract could still effectively reduce its purple pigment production by 27%, which proved that the crude extract had strong QSI activity.

### 3.3. Effect of Crude Extract on P. aeruginosa Quorum Sensing Inhibition

Both swarming and swimming motility are flagellum-dependent forms of movement observed in *P. aeruginosa*. The difference between swarming and swimming in *P. aeruginosa* is that swarming is the movement of a group of bacteria, whereas swimming is an individual motility behavior, both of which play a key role in the biofilm production of the strain in the later stage. The crude extract was added to the swarming and swimming agar medium at a final concentration of 5 mg/mL crude extract, respectively, to observe its effect on *P. aeruginosa* swarming and swimming. The results showed that *P. aeruginosa* swarming and swimming motility on agar plates without crude extracts were significant (Figure 5A,C), while its swarming and swimming motility were significantly inhibited by adding crude extract (Figure 5B,D).

Biofilms could reduce the penetration of antibacterial drugs into the cells, thereby reducing drug efficacy, which is one of the important causes of *P. aeruginosa* infection. The crude extracts of each concentration that did not significantly inhibit the growth of the strains were used to analyze the biofilm production of the strains. The results are shown in Figure 6; compared with the control group, all concentrations of crude extracts had a significant effect on the biofilm production of *P. aeruginosa*, and the biofilm inhibitory effect was gradually enhanced with the increase in crude extract concentration. The inhibition by *P. aeruginosa* biofilm was most significant at a concentration of 5 mg/mL, with 53% inhibition (Figure 6).

Pyocyanin is one of the important virulence factors and markers in quorum sensing of *P. aeruginosa*. Figure 7 lists the results of the effect of different concentrations of crude extract on the pyocyanin production of *P. aeruginosa*. There was a slight fall in pyocyanin production of *P. aeruginosa* with the increasing concentration of crude extract, and the inhibition rate ranged from 12.3% to 40.9%. The inhibition rate was more significant when the concentration was 5 mg/mL, and the inhibition rate was up to 40.9% (Figure 7), and the inhibition of pyocyanin production by *P. aeruginosa* was not significant at lower concentrations.

Rhamnolipids can indirectly enhance pathogenic activity by enhancing the communication ability between *P. aeruginosa*. Different concentrations of crude extracts were added to the medium to observe the effect on the production of rhamnolipids by *P. aeruginosa*. There is a clear trend of decreasing production of rhamnolipids by *P. aeruginosa* with an increase in the concentration of crude extracts, as shown in Figure 8. The inhibitory effect was most significant at 5 mg/mL and 2 mg/mL concentrations, with 27% and 19% inhibition, respectively, whereas the inhibition was not significant when the concentration was lower than that.

Proteases secreted by *P. aeruginosa* could degrade proteins and free amino acids in vitro, as well as create a variety of volatile and irritating compounds containing nitrogen and sulfur, which facilitate the invasion and spread of *P. aeruginosa* to the outside world. Therefore, the effect on the proteases secreted by *P. aeruginosa* was observed when different concentrations of crude extracts were used in this experiment. The protease inhibition rates were all around 6%, none of which produced significant inhibition (Figure 9).

### 3.4. Preliminary Analysis of Active Substances in Crude Extracts

The total ion chromatogram (TIC) of the crude extract is shown in Figure 10. 618 metabolic compounds were identified, and the data were uploaded to the National Microbial Science Data Center under the number NMDCX0000183. Thirty compounds were selected for functional validation after an analysis of their main compound structure types and related literature. The results showed that 19 of the selected compounds had significant QS inhibition activity at a concentration of 10 mg/L, and 8 of them also had more significant bacteriostatic inhibition activity (Table 2). Eugenol and 4-Allylphenol showed the highest inhibition activity with a QS area of 16.59 cm^2^ and 10.93 cm^2^, respectively, and also showed good Bacteriostatic activity with an inhibition area of 2.27 cm^2^. Isoeugenol also had good QS inhibition activity with an area of 6.5 cm^2^, while the other compounds showed relatively low QS inhibition activity.

## 4. Discussion

Since Fleming discovered penicillin in 1929, Actinomycetes have been an important source of natural active metabolites. According to incomplete statistics, there are 33,500 natural active products of microbial origin. Of these, 13,700 are of actinomycete origin (about 41%), while those of *Streptomyces* spp. account for about 80% of them [25]. Extreme desert environments are characterized by extreme aridity, intense solar ultraviolet radiation, and extreme temperature changes in the diurnal cycle, which breed a rich variety of extreme environmental microorganisms, of which actinomycetes are one of the most important components [26]. The Kumutag Desert, located in the eastern part of Xinjiang and south of Shanshan County, is part of the Taklamakan Desert. Its ecological environment is extremely special because of its hot and dry summers, with maximum temperatures reaching 49.6 °C and surface temperatures up to 82.5 °C, while the general minimum temperature in winter is −10 °C and extremely low temperatures can reach −29 °C, with occasional rain and snow throughout the year. Our group conducted preliminary isolation of microbial resources in the region and found that actinomycetes in the region accounted for the absolute dominance of the cultivable microbiota; we screened a strain with obvious QSI activity strain D67, and it shared the highest similarity with *Streptomyces ardesiacus* NBRC 15402T (98.39%) after preliminary analysis. According to the existing principles relating to the determination of new species [27], when the similarity between the 16S rRNA gene sequences of the two strains is lower than 98.65%, they can be judged to belong to different species. Therefore, it is tentatively considered a potential novel species in the *Streptomyces* genus and tentatively named *Streptomyces* sp. D67.

QSI as a new strategy for bacterial inhibition research has various sources, including animals, plants, and microorganisms, among which several actinomycetes have been reported to have QSI activity in their metabolites, such as *Streptomyces griseorubens*, *Nocardiopsis dassonvillei*, *Streptomyces parvulus*, and *Streptomyces xanthocidicus* [28,29]. Among them, there are many strains with QSI activity screened from the *Streptomyces* genus. Hassan et al. [30] found that 1H-pyrrole-2-carboxylic acid was a promising QSI natural product isolated from the soil-dwelling bacteria *Streptomyces coelicoflavus*, which could inhibit the expression of QS-related genes. Ooka et al. [31] isolated *Streptomyces phaeofaciens* from the soil, whose various metabolites, Piericidin A1, 3′-rhamnopiericidin A1, and a novel compound, piericidin E, had QSI activity. Zhou et al. [32] and Ishaque et al. [33] isolated *Streptomyces parvulus* and *Streptomyces tendae* from the surface seawater in the Lianyungang Sea area and the soil of the Rameswaram coast in southern India, respectively, and their products had QSI activity. The metabolite penicillin acylase of *Streptomyces lavendulae* isolated by Velasco-Bucheli et al. can efficiently and effectively hydrolyze the amide bonds of several AHLs [34]. The strain *Streptomyces* sp. D67 obtained in this study is a potential novel species in *Streptomyces*, but it shared a high similarity to both reported *Streptomyces coelicoflavus* and *Streptomyces parvulus* with 97.89% and 97.74%, respectively, predicting that strain D67 may have similar QSI activity products or metabolic pathways as the two strains, and further research is needed.

Currently, the QS network of *P. aeruginosa* is the most intensively studied, and there are relatively more relevant QSIs. It has been found that several pathways can inhibit QS in *P. aeruginosa*, such as m-bromothiolide, a signal molecule analogue, which can compete with the signal molecule for the RhlR receptor, thereby inhibiting the production of pyocyanin and biofilm formation [35]. Fermentation extracts of *Rhizobium* sp. containing AHL analogues (especially C4-HSL analogs) significantly reduce the production of biofilm and virulence factors in *P. aeruginosa* [36]; Park et al. [37] obtained an AHL-degrading enzyme from *Streptomyces* sp. M664, while *Bacillus pumilus* specifically degrades 3-oxo-C12-HSL, thus cutting off the quorum sensing in strains [38]. Meanwhile, many compounds from different sources, such as ajoene, tea polyphenols, curcumin, sphingocynoic acid, and myristic acid, have been shown to inhibit QS in *P. aeruginosa* [39,40,41,42,43]. In this study, preliminary analysis of the metabolic crude extract of *Streptomyces* sp. D67 was performed, and it revealed that the crude extract had a strong QS inhibitory activity against *C. violaceum* CV026 and significantly inhibited the swarming and swimming of *P. aeruginosa*. It also displayed significant inhibitory activity on the production of biofilm, pyocyanin, and rhamnolipid in *P. aeruginosa* in a significant concentration-dependent manner, and the highest inhibition rates were 53%, 40.9%, and 27%, respectively, but the inhibition effect on protease was not obvious. Related studies demonstrated that QscR in *P. aeruginosa* is an orphan receptor that has time-dependent regulation of the gene *lasB*, which encodes a protease, allowing the repression of the gene to persist into the early stationary phase [44]. Therefore, it is speculated that the crude extract of strain D67 may not inhibit QscR significantly or that its inhibition is time-limited. Meanwhile, the high concentration of crude extract of strain D67 showed significant bacterial inhibition activity, which suggested that the fermentation broth of strain D67 could inhibit the growth of harmful bacteria through a combination of QS inhibition and bacterial inhibition activity.

After preliminary analysis of the metabolites in the fermentation crude extract of *Streptomyces* sp. D67 by LC-MS, 19 compounds were found to have quorum-sensing inhibitory activity, including the already reported cinnamic acid, baicalein, and coumarin [45,46,47], while the QS inhibitory activity of 9-amino-1,2,3,4-tetrahydroacridine, p-allylphenol, 3,4-Dihydroxyhydrocinnamic acid, and 3-(4-hydroxyphenyl) lactic acid was rarely reported. In addition, eight compounds, such as cinnamaldehyde and eugenol, were also found to have more significant antibacterial activity at the same time. The above results demonstrated the metabolic capacity of compounds that inhibit QS activity in *Streptomyces* D67, which provides a theoretical basis for its application in food processing and plant biocontrol. Functional evaluation and fermentation metabolism optimization of related compounds are to be pursued to further solidify the authenticity of the related compounds.

## Figures and Tables

**Figure 1 microorganisms-11-02079-f001:**
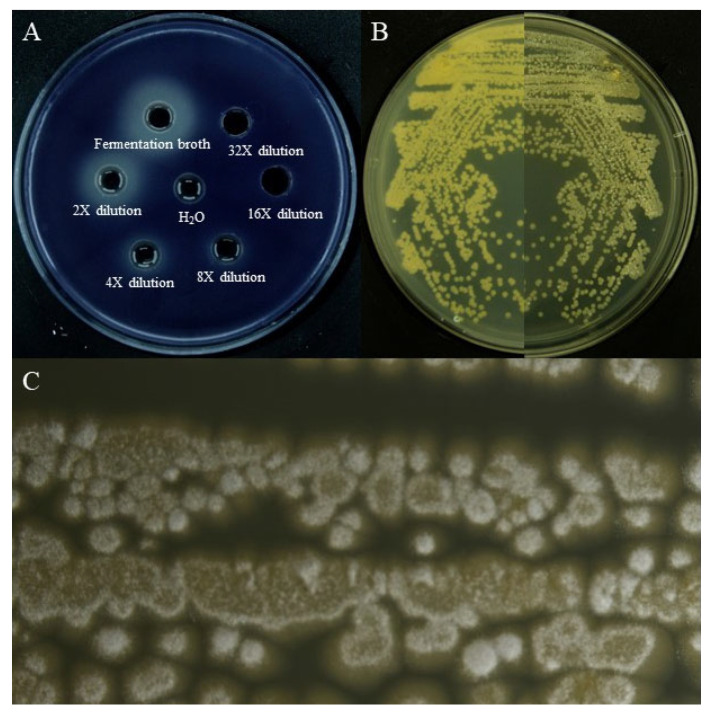
Screening and colony morphology of strain D67. (**A**) Inhibitory effect of strain D67 fermentation broth in different dilution gradients on the purple pigment of *C. violaceum* CV026; (**B**) Colony morphology of strain D67; (**C**) Colony micromorphology of strain D67. The Nikon NikonSMZ25 automatic microscope was used, the objective lens was 2×, the conversion magnification was 0.63×, and the overall magnification was 1.26×.

**Figure 2 microorganisms-11-02079-f002:**
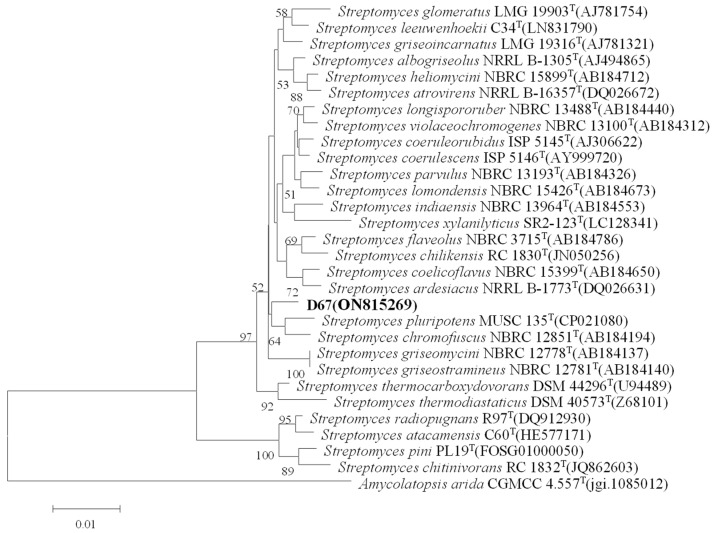
Neighbor-Joining tree based on 16S rRNA gene sequences showing the phylogenetic relationships among strain D67 and other related strains. Bootstrap values (≥50%) based on 1000 replications are shown at branch nodes. Bar 0.01 sequence variation.

**Figure 3 microorganisms-11-02079-f003:**
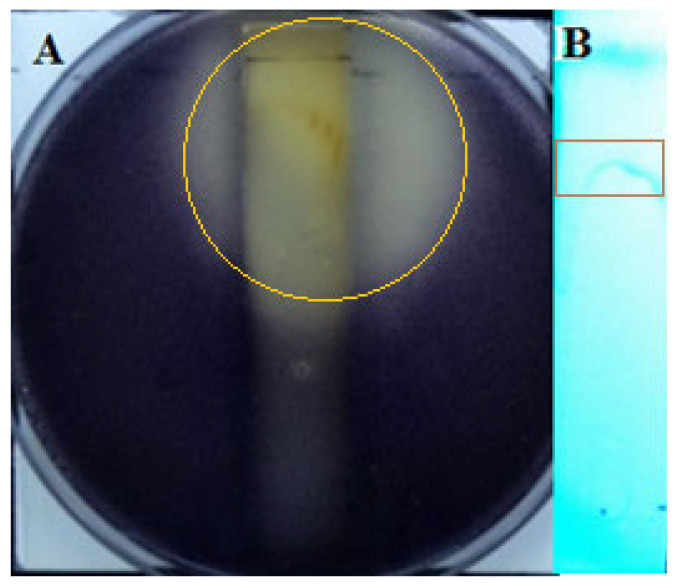
Analysis of crude extracts of strain D67 by bioautography of active ingredients and TLC. (**A**) Bioautography of active components in crude extract; (**B**) TLC of crude extract on 254 nm.

**Figure 4 microorganisms-11-02079-f004:**
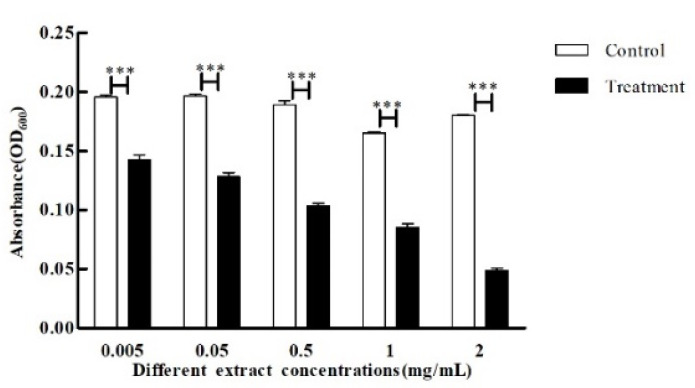
Effect of crude extracts with different concentrations on purple pigment production of CV026. The standard deviations of the biological replicates are represented by error bars and the mean ± SD from three independent experiments are shown. All the data were subjected to statistical analysis with a Duncan test to calculate the *p* value. ***, *p* < 0.001.

**Figure 5 microorganisms-11-02079-f005:**
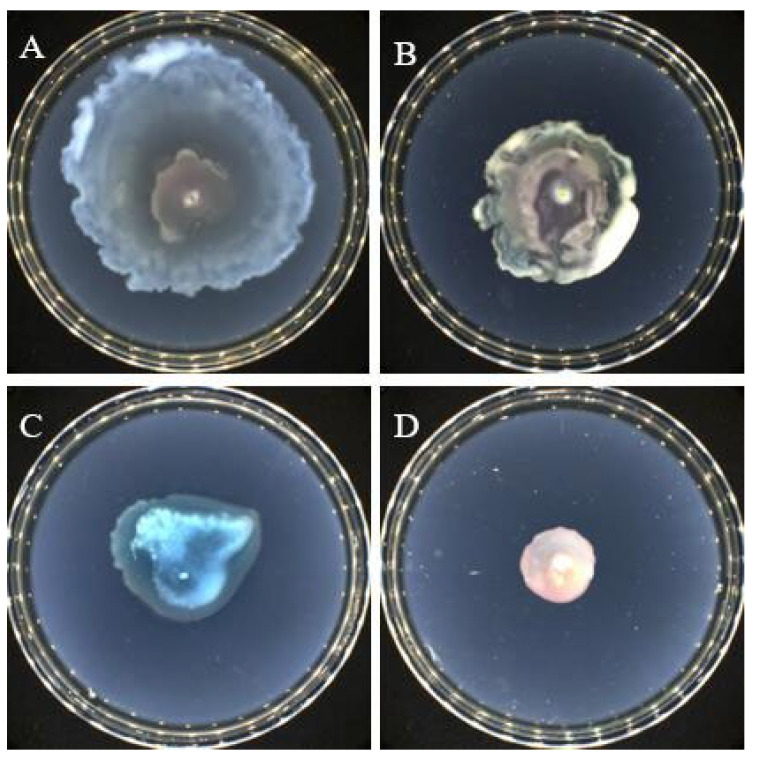
Effect of crude extracts (5 mg/mL) on swarming and swimming of *P. aeruginosa*. (**A**) Swarming control group; (**B**) Swarming experiment group (with 5 mg/mL crude extract); (**C**) Swimming control group; (**D**) Swimming experiment group (with 5 mg/mL crude extract).

**Figure 6 microorganisms-11-02079-f006:**
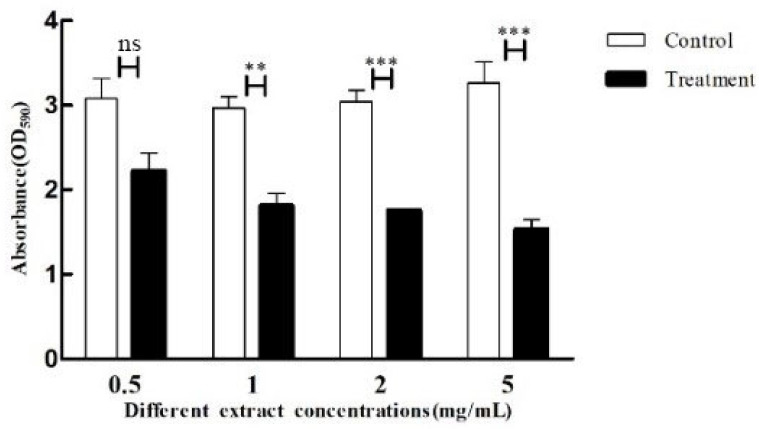
Effect of crude extracts with different concentrations on biofilm production of *P. aeruginosa*. The standard deviations of the biological replicates are represented by error bars and the mean ± SD from three independent experiments are shown. All the data were subjected to statistical analysis with a Duncan test to calculate the *p* value. ***, *p* < 0.001; **, *p* < 0.01; ns, *p* > 0.05.

**Figure 7 microorganisms-11-02079-f007:**
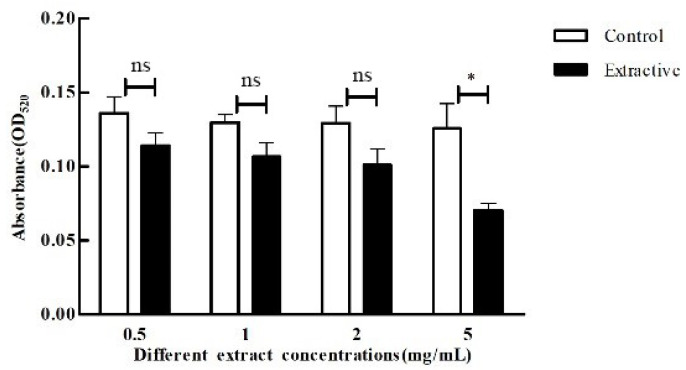
Effect of crude extracts with different concentrations on pyocyanin production of *P. aeruginosa*. The standard deviations of the biological replicates are represented by error bars and the mean ± SD from three independent experiments are shown. All the data were subjected to statistical analysis with a Duncan test to calculate the *p* value. *, *p* < 0.05; ns, *p* > 0.05.

**Figure 8 microorganisms-11-02079-f008:**
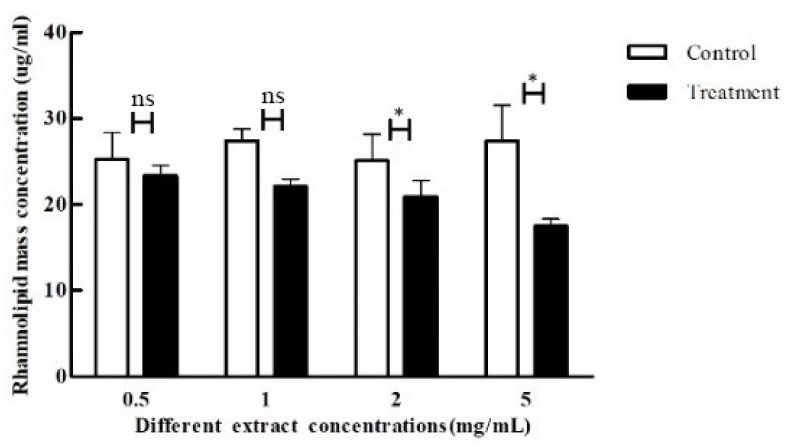
Effect of crude extracts with different concentrations on rhamnolipid production of *P. aeruginosa*. The standard deviations of the biological replicates are represented by error bars and the mean ± SD from three independent experiments are shown. All the data were subjected to statistical analysis with a Duncan test to calculate the *p* value. *, *p* < 0.05; ns, *p >* 0.05.

**Figure 9 microorganisms-11-02079-f009:**
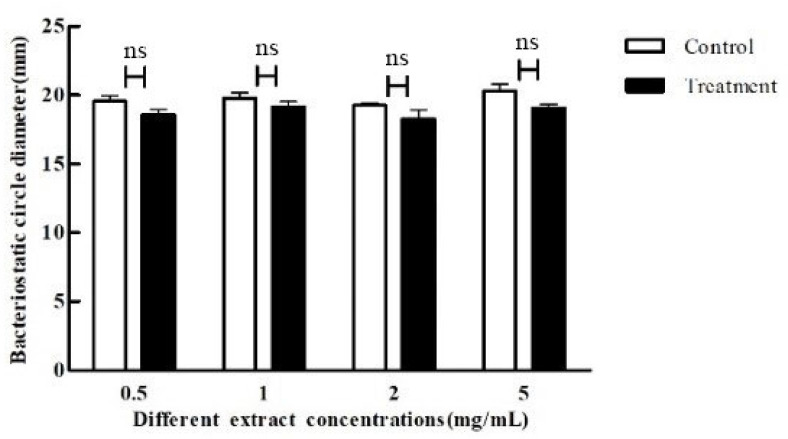
Effect of crude extracts with different concentrations on protease production by *p. aeruginosa*. The standard deviations of the biological replicates are represented by error bars and the mean ± SD from three independent experiments are shown. All the data were subjected to statistical analysis with a Duncan test to calculate the *p* value; ns, *p* > 0.05.

**Figure 10 microorganisms-11-02079-f010:**
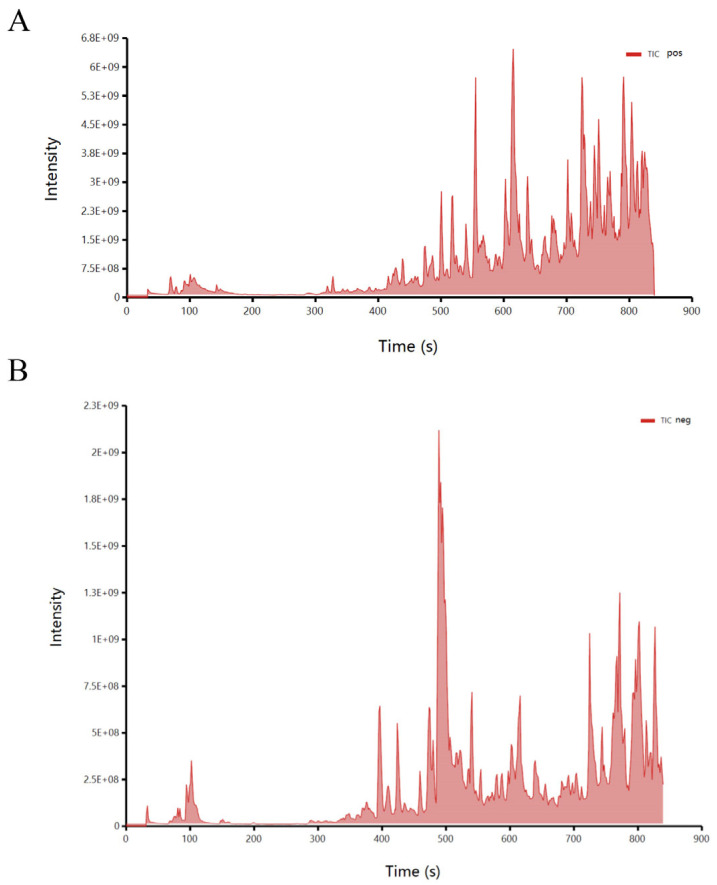
Total ionic chromatogram of crude extract of strain D67. (**A**) positive ionic chromatogram of crude extract; (**B**) negative ionic chromatogram of crude extract.

**Table 1 microorganisms-11-02079-t001:** Effect of crude extracts with different concentrations on the growth of strains.

Concentration of Crude Extracts (mg/mL)	*C. violaceum* CV026	*P. aeruginosa*
20	−	−
10	−	W
5	W	+
2	+	+
1	+	+
0.5	+	+
0.05	+	+
0.005	+	+

Note: −: the strain does not grow; +: Normal growth of the strain; W: Strain grows slowly.

**Table 2 microorganisms-11-02079-t002:** Quorum sensing inhibitory activity of 19 compounds.

Compound	Inhibition Area (cm^2^)	Violacein Inhibition Area (cm^2^)
Eugenol	2.27	16.59
4-Allylphenol	2.27	10.93
Isoeugenol	1.54	6.5
Cinnamaldehyde	2.84	6.24
Methyl 3-formylbenzoate	-	4.52
Caffeine	-	4.15
Cinnamic acid	-	4.15
1,2,3,4-Tetrahydroa cridin-9-amine	2.54	3.62
trans-o-Coumaric acid	-	3.46
Baicalein	-	3.14
Vanillic acid	-	3.14
Coniferyl alcohol	2.54	2.77
3-Indolylacetic acid	-	2.54
(4-Ethoxyphenyl)urea	-	2.54
6-Gingerol	-	1.77
Pyrrole-2-carboxylic acid	0.2	1.57
3,4-Dihydroxyhydro cinnamic acid	0.2	1.57
Coumarin	-	1.54
DL-4-Hydroxyphenyllactic acid	-	0.79

Note: -: the strain does not grow.

## Data Availability

Not applicable.

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
