# Peer review of "Screening and Identification of a Streptomyces Strain with Quorum-Sensing Inhibitory Activity and Effect of the Crude Extracts on Virulence Factors of Pseudomonas aeruginosa"

_microorganisms, 2023, doi:10.3390/microorganisms11082079_

Round 1

Reviewer 1 Report

In this paper, the authors screened and examined the inhibitory effects of bacteria on quorum-sensing. They found that the crude extracts of a Streptomyces showed the inhibitory activity and also showed some other biological activities on virulence factors. I have some comments and questions for the paper.

Line 2, “Streptomyces” needs to be italic.

Line 21-22, it’s kind of ambiguous for the description of “The fermentation crude extracts of strain D67 can effectively reduce the violacein produced...”. What is reduced here, production or activity of violacein? This needs to be clear.

Line 87, “P. aeruginosa” italic.

Figure 1A, I suppose that different circles represent different dilution gradients? It needs to be labeled in the figure or explained in the figure legend.

Figure 4, what do the error bars mean here? SD, SEM or other? The number of independent experiments needs to be mentioned in figure legend. What specific statistical analysis was used to calculate P value here? Same questions for the bar graphs and P values in Figures 6, 7, 8, 9.

Figure 6, statical analysis seems missing for 0.5 mg/ml. If there is no significant difference, I suggest using “ns” or “no significance”, otherwise it’s confusing. Same as the bar graphs in other Figures.

Line 319, how were the 618 metabolic compounds identified? More details need to be given.

Figure 10, do the authors have media control for the chromatogram of crude extract?

Table 2, I guess it’s only 1-time experiment for the inhibitory activity measurements of 19 compounds, is that correct? There are no +/- values here.

Reviewer 2 Report

The present article entitled "Screening and identification of a Streptomyces strain with quorum-sensing inhibitory activity and effect of the crude extracts on virulence factors of Pseudomonas aeruginosa" based on a good theme and authors have designed and well presented the article.

However article need some corrections and clarification like:

-Make the strains name italic in most of the place authors have not make the name in italic

-Authors have mentioned  isolation of  Streptomyces from the soil samples, but after sampling  various microbial strains will be appeared on the plate, but on what basis authors have selected  the mentioned strains for further study: add the details 

The present article entitled "Screening and identification of a Streptomyces strain with quorum-sensing inhibitory activity and effect of the crude extracts on virulence factors of Pseudomonas aeruginosa" based on a good theme and authors have designed and well presented the article.

However article need some corrections and clarification like:

-Make the strains name italic in most of the place authors have not make the name in italic

-Authors have mentioned  isolation of  Streptomyces from the soil samples, but after sampling  various microbial strains will be appeared on the plate, but on what basis authors have selected  the mentioned strains for further study: add the details 

Round 2

Reviewer 1 Report

I'm ok with the revised version. Thank you for addressing my concerns.